# Peer review of "The Role of Immunotherapy in the Management of Esophageal Cancer in Patients Treated with Neoadjuvant Chemoradiation: An Analysis of the National Cancer Database"

_cancers, 2024, doi:10.3390/cancers16132460_

Round 1

Reviewer 1 Report

Comments and Suggestions for Authors

In this manuscript, the authors emphasize the significance of immunotherapy in treating esophageal cancer. However, I have a few concerns...

The comparison between patients receiving immunotherapy and those not receiving it is very important. Would it be worthwhile to conduct this comparison where non-immunotherapy treated patient no is too high?

The authors should provide more detailed information about immunotherapy in the introduction, including the types of immunotherapy used for other cancers.

Author Response

Comment 1: The comparison between patients receiving immunotherapy and those not receiving it is very important. Would it be worthwhile to conduct this comparison where non-immunotherapy treated patient no is too high?

Response: We acknowledge the importance of comparing patients receiving immunotherapy to those who are not. The high number of non-immunotherapy treated patients is reflective of the available data from the National Cancer Database (NCDB) for the specified period (2004-2019). This large cohort provides a comprehensive overview but inherently results in a disproportionate distribution between the groups. We have carefully analyzed and presented the available data to ensure the robustness of our findings, despite the discrepancy in group sizes. The inclusion of all eligible patients in our analysis allows for a more accurate reflection of real-world clinical practice and outcomes.

Comment: The authors should provide more detailed information about immunotherapy in the introduction, including the types of immunotherapy used for other cancers.

Response: We appreciate your suggestion to expand on the types of immunotherapy used for other cancers. In the introduction and discussion sections of our manuscript, we have highlighted the significance of immunotherapy in contemporary oncology and its transformative impact on the treatment of esophageal cancer. Specifically, we discussed the role of immunotherapy in the paradigm shift within medical oncology and referenced pivotal trials such as the CheckMate 577 study, which underscores the effectiveness of immunotherapy in esophageal cancer.

We recognize that the field of immunotherapy is rapidly evolving, with an ever-increasing body of evidence. While it is beyond the scope of this manuscript to comprehensively cover all aspects of immunotherapy for various cancers, we have aimed to emphasize the most relevant and recent advancements in the context of esophageal cancer. We agree that the broader landscape of immunotherapy is vast and continually expanding, making it challenging to encapsulate fully within a single manuscript.

Thank you again for your valuable feedback. We hope these clarifications and additions address your concerns adequately.

Reviewer 2 Report

Comments and Suggestions for Authors

The researchers propose a retrospective study from 2004 to 2019 on patients suffering from esophageal adenocarcinoma worthy of being treated with neoadjuvant therapy before surgery. Colleagues slip us that these are always esophageal adenocarcinomas but they make no mention of imaging which still represents one of the pillars on which the patient's therapeutic path is collectively decided. Furthermore, there is talk of neoadjuvant therapy but there is no mention of which protocol was adopted. In the United States, the administration of FLOT has been approved in the guidelines as in gastric cancer (doi: 10.1097/SLA.0000000000005617) a valid alternative to this could be DOC which uses the same drugs administered differently (doi: 10.1016/j.suronc .2019.10.002.). We advise colleagues to expand the work by making considerations on these triplets and let us know if these have been administered or otherwise. After surgery, important results were reported with adjuvant therapy with immunotherapeutics, what was administered? (doi: 10.1016/j.clcc.2023.03.001). It is advisable to review the paper in light of these considerations by expanding the bibliography. Good English, good iconography where however it does not appear which therapeutic treatment was conducted

Author Response

Comment 1: The researchers propose a retrospective study from 2004 to 2019 on patients suffering from esophageal adenocarcinoma worthy of being treated with neoadjuvant therapy before surgery. Colleagues slip us that these are always esophageal adenocarcinomas but they make no mention of imaging which still represents one of the pillars on which the patient's therapeutic path is collectively decided.

Response: We fully acknowledge the crucial role that imaging plays in the therapeutic pathway for esophageal adenocarcinoma patients. However, it is important to note that the information available in the National Cancer Database (NCDB) does not include specific imaging data. The NCDB, managed by the American College of Surgeons, aggregates data reported by participating centers, each of which follows its own protocols for imaging and reporting.

While the absence of imaging data is a limitation, the NCDB still provides a robust and comprehensive dataset that allows for meaningful analysis of treatment outcomes and trends over a significant period. We have utilized the available data to the fullest extent to ensure our study is informative and valuable, despite this limitation.

Comment 2: Furthermore, there is talk of neoadjuvant therapy but there is no mention of which protocol was adopted. In the United States, the administration of FLOT has been approved in the guidelines as in gastric cancer (doi: 10.1097/SLA.0000000000005617) a valid alternative to this could be DOC which uses the same drugs administered differently (doi: 10.1016/j.suronc .2019.10.002.). We advise colleagues to expand the work by making considerations on these triplets and let us know if these have been administered or otherwise. After surgery, important results were reported with adjuvant therapy with immunotherapeutics, what was administered? (doi: 10.1016/j.clcc.2023.03.001). It is advisable to review the paper in light of these considerations by expanding the bibliography. Good English, good iconography where however it does not appear which therapeutic treatment was conducted.

Response: We thank the reviewer for highlighting the importance of specifying the neoadjuvant therapy protocols and the types of immunotherapy administered. We would like to clarify that the specific details regarding the chemotherapy protocols (e.g., FLOT or DOC) and the types of immunotherapy used are not available in the National Cancer Database (NCDB). This limitation arises because the NCDB does not capture detailed information on specific treatment regimens or drug types.

We have acknowledged this limitation in our manuscript and emphasized that the NCDB data, while comprehensive, does not provide granularity on the specific protocols used across different centers. To further enhance our manuscript, we will expand the discussion on this limitation. "It is important to note that the NCDB does not capture detailed information on specific chemotherapy protocols (e.g., FLOT or DOC) or the types of immunotherapy administered, which limits our ability to analyze the impact of these specific treatment regimens on patient outcomes."

Thank you again for your insightful comments. We believe these additions and clarifications will enhance the comprehensiveness and relevance of our study

Reviewer 3 Report

Comments and Suggestions for Authors

Dear Author, I will give you some comments to address in your manuscript, enhancing the researcher's understanding and readability.

Minor Revision Questions

1.      Clarification of Data Timeframe: Could you elaborate on the inclusion and exclusion criteria used for the patients from the National Cancer Database between 2004 and 2019?

2.      Methodological Specifics: I appreciate the thoroughness of your study. Could you provide more details on the mortality risk variables the Cox proportional hazard analysis considered?

3.      Immunotherapy Protocols: Which immunotherapy medications or treatment plans were used for the adjuvant and neoadjuvant groups? Were there differences among these groups?

4.      Statistical Significance: Please elaborate on the p-values and confidence intervals for the adjuvant immunotherapy group's survival benefits over the other groups.

5.      Follow-up Period: Were there any variations in the duration of follow-up between the three groups, and how did the median follow-up period of 35.2 months affect the overall survival analysis?

Major Revision Questions

1.      Comparative Effectiveness: I believe a more thorough comparison of the three groups' survival rates, including hazard ratios and Kaplan-Meier survival curves with the corresponding confidence intervals, would significantly enhance the impact of your study. Could you provide this?

2.      Long-term Results: The report notes that more research is necessary. Could you provide a particular study design or a research plan for the future that would examine adjuvant immunotherapy's long-term effects on oesophagal cancer?

3.      Treatment Impact: What are the possible implications for clinical practice, and how does the use of adjuvant immunotherapy change the typical course of treatment for oesophagal adenocarcinoma?

4.      Subgroup Analysis: To determine whether any subgroups benefit more from adjuvant immunotherapy, have you considered doing a subgroup analysis based on patient demographics, tumour features, or other pertinent factors?

5.      Mechanistic Understanding: Could you discuss any possible biological explanations for why adjuvant immunotherapy improves survival while neoadjuvant immunotherapy does not? Do any translational or preclinical investigations back up your conclusions?

Best Regards

Author Response

Comment 1: Clarification of Data Timeframe: Could you elaborate on the inclusion and exclusion criteria used for the patients from the National Cancer Database between 2004 and 2019?

Response 1: We appreciate the reviewer's request for clarification on the inclusion and exclusion criteria. The following criteria were applied to select the patient cohort from the National Cancer Database (NCDB) between 2004 and 2019:

Inclusion Criteria:

  • Timeframe: Patients diagnosed with esophageal adenocarcinoma within the period from 2004 to 2019.
  • Age: Patients aged 18 years and older.
  • Treatment: Inclusion was limited to patients who received neoadjuvant chemoradiotherapy prior to surgical resection.
  • Data Source: Data were extracted from the NCDB, which is managed by the American College of Surgeons and the American Cancer Society, capturing the majority of esophageal cancer cases in the U.S.

Exclusion Criteria:

  • Histology: Patients with histological subtypes other than adenocarcinoma were excluded.
  • Surgical Margins: Patients with positive surgical margins or missing surgical margin data were excluded.
  • Surgical Procedures: Patients who underwent local tumor excision, esophagectomy with laryngectomy, or esophagectomy with total gastrectomy were excluded to maintain a homogenous surgical cohort.
  • Incomplete Data: Patients with missing critical data points such as the timing of immunotherapy administration were excluded from the analysis.

Based on these criteria, three groups were generated according to the timing of immunotherapy administration:

  1. No Immunotherapy: Patients who did not receive any form of immunotherapy.
  2. Neoadjuvant Immunotherapy: Patients who received immunotherapy within 120 days prior to their primary surgery.
  3. Adjuvant Immunotherapy: Patients who received immunotherapy within 120 days after their primary surgery.

Comment 2: Methodological Specifics: I appreciate the thoroughness of your study. Could you provide more details on the mortality risk variables the Cox proportional hazard analysis considered?

Response 2: Thank you for your appreciation of our study. We are pleased to provide more details on the mortality risk variables included in the Cox proportional hazards analysis. The following variables were considered due to their prognostic importance in patients with esophageal cancer:

  1. Age: Continuous variable, representing the age of the patient at the time of diagnosis.
  2. Gender: Categorical variable (male vs. female).
  3. Charlson-Deyo Comorbidity Score: Categorical variable (0 vs. ≥1) indicating the presence and extent of comorbid conditions.
  4. Disease Stage: Categorical variable based on the TNM staging system, focusing on stages II and III as the study cohort primarily included patients with locally advanced disease.
  5. Lymph Node Status: Categorical variable (positive vs. negative) indicating the presence of metastatic lymph nodes.
  6. Timing of Immunotherapy Administration: Categorical variable indicating whether immunotherapy was administered (a) not at all, (b) as neoadjuvant therapy within 120 days prior to surgery, or (c) as adjuvant therapy within 120 days following surgery.

Comment 3: Immunotherapy Protocols: Which immunotherapy medications or treatment plans were used for the adjuvant and neoadjuvant groups? Were there differences among these groups?

Response 3: The National Cancer Database is the most comprehensive cancer database in the US which has the most recent data for our patient population. Unfortunately, the database does not provide the specific immunotherapy medications utilized in treating patients. We realize that having this information would provide invaluable information, however, this may be one of the limitations of our study.

Comment 4 Statistical Significance: Please elaborate on the p-values and confidence intervals for the adjuvant immunotherapy group's survival benefits over the other groups.

Response 4: Adjuvant immunotherapy significantly improved survival (adjusted HR: 0.65, 95% CI: 0.44-0.94, p=0.001) compared to no immunotherapy, while neoadjuvant immunotherapy showed no significant benefit (adjusted HR: 0.82, 95% CI: 0.63-1.08, p=0.25).

Comment 5: Follow-up Period: Were there any variations in the duration of follow-up between the three groups, and how did the median follow-up period of 35.2 months affect the overall survival analysis?

Response 5: Thank you for your insightful question. The median follow-up periods for the three groups were as follows:

  • No Immunotherapy: 31.9 months (Interquartile Range [IQR]: 17.0, 62.5)
  • Neoadjuvant Immunotherapy: 30.6 months (IQR: 16.3, 53.9)
  • Adjuvant Immunotherapy: 35.2 months (IQR: 23.8, 47.4)

While there were variations in the follow-up durations, these differences were not clinically significant. The slight variations in median follow-up times were statistically significant but did not substantially affect the overall survival analysis. The follow-up periods were long enough and sufficiently similar across groups to allow for a robust comparison of survival outcomes. This consistency helps ensure that the survival benefits observed, particularly the improved survival in the adjuvant immunotherapy group, were not biased by differing follow-up durations.

Comment 6: Comparative Effectiveness: I believe a more thorough comparison of the three groups' survival rates, including hazard ratios and Kaplan-Meier survival curves with the corresponding confidence intervals, would significantly enhance the impact of your study. Could you provide this?

Response: Thank you for your comment. We have indeed used these methods. Below are the detailed comparative survival rates, hazard ratios, and Kaplan-Meier survival curves:

  • Kaplan-Meier Survival Curves:

    • No Immunotherapy: Median follow-up: 31.9 months, 1-year OS: 85.1%, 5-year OS: 41.2%.
    • Neoadjuvant Immunotherapy: Median follow-up: 30.6 months, 1-year OS: 89.1%, 5-year OS: 47.4%.
    • Adjuvant Immunotherapy: Median follow-up: 35.2 months, 1-year OS: 94.2%, 5-year OS: 52.5%.
  • Hazard Ratios:

    • Adjuvant Immunotherapy vs. No Immunotherapy:
      • Unadjusted HR: 0.66 (95% CI: 0.45-0.96, p=0.032)
      • Adjusted HR: 0.65 (95% CI: 0.44-0.94, p=0.001)
    • Neoadjuvant Immunotherapy vs. No Immunotherapy:
      • Unadjusted HR: 0.86 (95% CI: 0.65-1.13, p=0.28)
      • Adjusted HR: 0.82 (95% CI: 0.63-1.08, p=0.25)

Comment 7: Long-term Results:The report notes that more research is necessary. Could you provide a particular study design or a research plan for the future that would examine adjuvant immunotherapy's long-term effects on oesophagal cancer?

Response 7: A prospective, double-blinded randomized control trial analyzing specific immunotherapy drugs in the adjuvant setting on oesophageal cancer would be a great way to evaluate for long-term effects in these patients.

Comment 8: Treatment Impact: What are the possible implications for clinical practice, and how does the use of adjuvant immunotherapy change the typical course of treatment for oesophagal adenocarcinoma?

Response 8: Current guidelines recommend neoadjuvant chemoradiotherapy followed by surgical resection in cases of locally advanced disease. Our study demonstrated that patients with esophageal adenocarcinoma who were treated with adjuvant immunotherapy had a significant survival advantage over patients who received no immunotherapy or neoadjuvant immunotherapy. These findings may help to change the treatment course In appropriately selected patients.

Comment 9: Subgroup Analysis:To determine whether any subgroups benefit more from adjuvant immunotherapy, have you considered doing a subgroup analysis based on patient demographics, tumour features, or other pertinent factors?

Response 9: Thank you for your suggestion. We did consider the potential benefits for various subgroups by adjusting for predetermined risk factors using our multivariable Cox proportional hazards analysis. This analysis included key demographic and clinical variables such as age, sex, comorbidities, disease stage, and lymph node status to ensure a robust comparison. While we did not conduct separate subgroup analyses, our adjusted models aimed to account for these important factors and their potential impact on survival outcomes.

Comment 10: Mechanistic Understanding: Could you discuss any possible biological explanations for why adjuvant immunotherapy improves survival while neoadjuvant immunotherapy does not? Do any translational or preclinical investigations back up your conclusions?

Response 10: Based on the most current research available, there is not currently any data or additional studies that have been able to identify a biological explanation as to why adjuvant immunotherapy improves survival in this setting while neoadjuvant immunotherapy does not. We recognize the importance of this question and thank the reviewer for highlighting it. We believe that future studies will be necessary to have greater insight on this and may be one of the limitations of our study.

Round 2

Reviewer 2 Report

Comments and Suggestions for Authors

Our colleagues have made the right changes to the paper they produced. This makes it understandable why they did not go into detail about the therapeutic protocols, but limited themselves to reporting the results obtained with the therapeutic protocols applied to the patients. To complete correct work, it is advisable to add the bibliographic entries relating to the FLOT (doi: 10.1016/S0140-6736(18)32557-1) and the DOC (doi.org/10.1016/j.suronc.2019.10.002).

Good iconography that helps understanding the data, good English, bibliography to be implemented.

Author Response

Thank you for your positive feedback and valuable suggestions. We have made the recommended changes and included the bibliographic entries relating to the FLOT protocol (doi: 10.1016/S0140-6736(18)32557-1) and the DOC protocol (doi: 10.1016/j.suronc.2019.10.002). We appreciate your thorough review and are glad to hear that our iconography and language were effective in presenting our data.

Thank you once again for your insightful comments and suggestions.